# Complexity Synchronization of Organ Networks

**DOI:** 10.3390/e25101393

**Published:** 2023-09-28

**Authors:** Bruce J. West, Paolo Grigolini, Scott E. Kerick, Piotr J. Franaszczuk, Korosh Mahmoodi

**Affiliations:** 1Department of Research and Innovaton, North Carolina State University, Raleigh, NC 27606, USA; 2Center for Nonlinear Science, University of North Texas, Denton, TX 76203, USA; 3US Combat Capabilities Command, Army Research Laboratory, Aberdeen Proving Ground, MD 21005, USA; 4Department of Neurology, Johns Hopkins University School of Medicine, Baltimore, MD 21287, USA

**Keywords:** complexity, synchronization, fractal coding, organ networks, complexity synchronization, scaling

## Abstract

The transdisciplinary nature of science as a whole became evident as the necessity for the complex nature of phenomena to explain social and life science, along with the physical sciences, blossomed into complexity theory and most recently into complexitysynchronization. This science motif is based on the scaling arising from the 1/f-variability in complex dynamic networks and the need for a network of networks to exchange information internally during intra-network dynamics and externally during inter-network dynamics. The measure of complexity adopted herein is the multifractal dimension of the crucial event time series generated by an organ network, and the difference in the multifractal dimensions of two organ networks quantifies the relative complexity between interacting complex networks. Information flows from dynamic networks at a higher level of complexity to those at lower levels of complexity, as summarized in the ‘complexity matching effect’, and the flow is maximally efficient when the complexities are equal. Herein, we use the scaling of empirical datasets from the brain, cardiovascular and respiratory networks to support the hypothesis that complexity synchronization occurs between scaling indices or equivalently with the matching of the time dependencies of the networks’ multifractal dimensions.

## 1. Introduction

The physical concept of *synchronization* is nearly four centuries old, whereas the idea of *complexity* being sufficiently broad to constitute a science on its own is less than four decades old [1]. In this paper, we follow in the tradition of interdisciplinary studies and propose conjoining two distinctly different empirical constructs into a single concept, that being *complexity synchronization*, with the intention of learning something new. It is remarkable that complexity synchronization does, in fact, define a new phenomenon, which in turn provides fresh insight into the health, disease and rehabilitation of living networks. Complexity science herein produces a simple rule that underpins the complexity of living networks and how this underpinning is achieved constitutes the theme of this paper:

Traditional science seeks direct causal relations between elements in the universe, whereas complexity theory drops down a level to explain the rules that govern the interactions between lower-order elements that in the aggregate create emergent properties in higher-level systems.[1]

In Computer Science, the concept of a distributed shared-memory network describes several computers that share a memory area, but because the variability in speed among computers has no global clock with which to order activities, network synchronization is introduced to maintain order [2]. The utility of this theoretical concept has been determined in many natural systems as well. In social sciences, synchronization has been identified as an empirical mechanism that coordinates activities between events within and between networks, but as networks become more complex so too does the concept of network synchronization. This is particularly true of the amazingly complex living network structure of the human body and the need to coordinate activities from the microscopic time scales of the chemical reactions within neural networks constituting the brain, to the mesoscopic time scales of the cardiac and respiratory networks, to the macroscopic time of circadian rhythms. For our purposes, here we intend to capture the complexity of living networks in our discussion by using the more focused and less controversial term organ network (ON).

The data processing approach revealing this new kind of synchronization is based on time series consisting of random discrete events whose statistics are of the renewal type and which enable the detection and quantification of synchrony among ONs operating on different time scales and not necessarily in stationary regimes [3]. The discrete events in such time series have been named crucial events [4] because they determine the efficiency of the information exchange between these complex ONs, and for a large class of complex networks, not necessarily only ONs, crucial events determine network failures—from heart attacks to stock market crashes.

Crucial event time series are generated by inverse power-law waiting-time probability density functions $\psi \left(t\right)\propto t^{-\mu }$ with an inverse power-law index in the domain 1<μ<3. Asymptotically, the generated crucial event time series describes an ergodic process for the inverse power-law index μ in the range 2<μ<3 with a finite average waiting time. Ergodic is the technical name for statistical processes for which averages taken over long time series are equal to those taken over probability density functions. The understanding of most of the complexity arising in many-body physics is understood using the ‘ergodic hypothesis’ dating back to Boltzmann.

A crucial event time series having the waiting-time inverse power-law index in the range 1<μ<2 results in an infinite average waiting time and is therefore non-ergodic. One of the simpler ways to determine whether a time series is non-ergodic is by noting whether a measurement at two times depends not on the time difference (stationary, ergodic) but rather on the two time points separately (non-stationary, non-ergodic). Consequently, in this range of μ<2, most of the mathematical infrastructure developed using the traditional many-body theory of physics cannot be transferred and new methods must be sought.

Herein, we show through the data processing of empirical time series that physiologic ONs generate crucial event time series, that is, the events have statistically independent time intervals and are therefore of the renewal type. In this paper, we focus on the empirical complexity of electroencephalographic (EEG) data being multifractal, as are the respiratory and cardiovascular time series, and establish that the three multifractal scalings are synchronous [3,5]. This remarkable synchrony among the three ONs’ time series is the empirical evidence for the existence of complexity synchronization, as well as its fundamental importance in coordinating the functions of various ONs for the healthy operation of the human body.

The multifractal behavior of these three time series has previously been identified using the pairwise correlation of time series to identify a synchronizing mechanism [6]. Note that the synchrony between two time series is not the same as the synchrony of the scaling parameters which occurs for criticality matching, the latter being a locking of the scaling indices in time and not necessarily a locking of the time series themselves. However, complexity synchronization does not require this lower-order synchrony in order for it to be the mechanism whereby body organs effectively communicate among themselves and thereby function as a cohesive whole.

The change in fractal dimensions, as determined by the different scaling indices of the time series, indicates the changing complexity of the ONs as various physiological functions are performed. For example, information is readily transported within overlapping memory areas of the heterogeneously complex brain, and at any point in time a given region of the brain may be able to receive from, or transmit information to, another physiological ON, depending on their function and instantaneous relative complexities. This ever-changing hierarchy of complexities is revealed herein by the way in which the multifractal nature of each of these three interacting ONs influence one another over time, as we subsequently show.

Complexity is one of those concepts that although often used in multiple disciplinary contexts eludes unique formal definition. So, to avoid becoming embroiled in a semantic debate, we herein adopt a working definition for complexity that appears to be more than satisfactory for describing ON-generated crucial event time series.

### 1.1. Working Definition of Complexity

A signal X(t) generated by an ON is given by a time sequence of crucial events, whose probability density function for the time interval between such events is an inverse power law [7]. The time series scales with a scaling index δ if for a given parameter λ we can write the homogeneous scaling relation $X\left(\lambda t\right)=\lambda {}^{\delta }X\left(t\right)$. It is readily determined that a signal’s level of complexity, as measured by the fractal dimension *D*, increases for the crucial event time series as the inverse power-law index μ increases. Consequently, the fractal dimension of a crucial event time series is given by the relation D=2−δ and increases with increasing complexity [8]. Table 1 records the scaling in the power spectral density, the probability density function and the dynamic variable denoting the time series, as well as the relations among the various scaling indices μ, β and δ defining the scaling properties of the crucial event time series.

It was hypothesized [7] and later proven [9,10] that the information flow between interacting networks depends on their relative complexity, then called the ‘complexity matching effect’. In a fashion analogous to the flow of energy following a negative energy gradient, the flow of information follows an information (negative entropy) gradient. Consequently, the information exchanged between two such interacting fractal ONs is maximally efficient when the two complexity measures given by their respective fractal dimensions are equal [7].

We show herein that at each instant of time a local complexity of the brain (as measured by the fractal dimension of a local EEG channel time series) is either tracked or driven by the complexity of the respiratory and cardiovascular ONs, with the relative complexity of these and other physiologic ONs being task dependent. Information is readily transported within the heterogeneously complex brain, as described above. This changing hierarchy of the local complexity is revealed herein by the way in which the multifractal nature of each of these three interacting ONs influences the other two ONs over time.

### 1.2. Multiple
Measures of Complexity

The previous paragraph might give the mistaken impression that because the power spectrum for a crucial event time series has a unique value for its inverse power-law index β, as does the scaling index δ, that the inverse power-law index for the waiting-time probability density function μ shares this property of uniqueness. It does not. There are, in fact, at least two ways to measure this last index.

The first way is to use the relation given in Table 1 and to assume that at a given time $t_{j}$ the trajectory X(t) has the value $X_{j}$. How much time do we wait before the trajectory takes on this value again? This recrossing probability density function is the waiting-time probability density function given in the table, which we will denote by replacing the generic inverse power-law index for the waiting-time probability density function μ by $\mu_{D}$. From the other parameter relations in Table 1, it is clear that
(1)$$\mu_{D}=2-\delta ,$$
where the scaling index is that given in the table. The crossing and recrossing of the trajectory at the fixed value of the diffusion process $X_{j}$ has been shown to be a renewal process [11]. To establish a connection between the waiting-time index $\mu_{D}$ with the scaling probability density function index given in the Methods Section, we address the specific cases of super-diffusion and sub-diffusion.

It is important to stress in the field of fractal dynamics [8] the relation between the fractal dimension *D* and the Hurst exponent *H*: D=2−H, which Mandelbrot and Van Ness [12], using the fractional calculus, interpreted to be a dynamic fractal process, which they named fractional Brownian motion (FBM). This suggests interpreting $\mu_{D}$ to be a fractal dimension by interpreting *H* as the scaling index δ, as is often done, thereby yielding the fractal dimension
(2)D=2−δ,
thereby establishing a general connection between the fractal dimension and scaling $D=\mu_{D}$. The argument given here attracts our attention to the field of complexity research by which we realize that FBM presents a singularity. Recall that Equation (Equation 2) was obtained using algebra alone from the relations recorded in Table 1, without the insight provided here.

The second way to obtain the inverse power-law index for the waiting-time probability density function is to examine the situation for the extremes of anomalous diffusion, those being super-diffusion (δ>0.5) and sub-diffusion (δ<0.5). The super-diffusion case is addressed using the crucial events described by the inverse power-law index denoted by $\mu_{S}$ replacing the generic index μ in Table 1. Using Equation (Equation 1), with the left side of the equation interpreted as the fractal dimension and the scaling index δ on the right side expressed in terms of $\mu_{S}<2$, yields $\mu_{D}=2-1/\left[\mu_{S}-1\right]$, which after rearranging the terms yields
(3)$$\mu_{S}=1+1/\left[2-\mu_{D}\right].$$
In order for the condition specified by this equation to be satisfied requires that between the consecutive crucial events in the signal driving the diffusion process is assumed either the value of +1 or −1 by means of an equal probability coin toss.

The case of sub-diffusion with δ<0.5 is addressed as performed by Failla et al. [11] by assuming $1<\mu_{S}<2$. The fluctuating driver of the diffusion process is assumed to vanish between consecutive crucial events and to take on the value of either +1 or −1 with equal probability at the time of a crucial event occurring. This yields the relation $\mu_{D}=2.5-\mu_{S}/2$ so that, again, rearranging the terms gives us
(4)$$\mu_{S}=5-2\mu_{D}.$$
It is important to notice the increasing interest in the emergence of $\mu_{S}$, heralding the non-ergodic behavioral dynamics along with criticality in the discussion of scale-free cortical dynamics [13].

The Self-Organized Temporal Criticality model spontaneously generates temporal complexity by means of the criticality of a network’s dynamics. The global fluctuations around the mean are calculated as in the inverse power-law probability density function. Of particular interest to us is the monitoring of the times at which the fluctuations cross the origin, giving rise to the first passage time power-law index [14] $\mu_{D}=1.3$. Inserting this value into Equation (Equation 4) yields $\mu_{S}=2.4$, which explains why the inverse power-law index lies in the interval $2<\mu_{S}<3$ and the crucial event time series is therefore ergodic as well as being of the renewal type. It is important to stress that the well-known model of criticality and complexity given by Vicek et al. [15] and characterized by Vanni et al. [16] yields essentially the same results.

In summary, the interpretation of $\mu_{D}$ as a fractal dimension is always correct. The interpretation of the fractal dimension of the time series *D* as the first passage time index $\mu_{D}$ as a renewal process does not apply to fractional Brownian motion. We stress that it is not easy to assess if a fractal process also satisfies the renewal condition, thereby emphasizing the significance of the results found for the triad of empirical time series studied herein.

### 1.3. Brief History of Synchronization

The first recorded articulation of the concept of synchronization, separate and distinct from that of complexity, was provided in a letter by the Dutch physicist Huygens in 1665. He observed what he called ‘the sympathy of two clocks’, wherein, despite independent initial conditions, the pendulums of two clocks hanging from the same beam synchronized in the anti-phase within thirty minutes. At this time, 22 years before Newton’s *Principia*, a consistent vocabulary of mechanical forces with which to understand the synchronization phenomenon did not yet exist. The concept of synchronization has since evolved from the physical similarity of two oscillatory time series to more abstract measures of similarity based on recent advances in the nonlinear dynamics of many-body networks. Thus, today’s notion of synchronization differs from Huygen’s original concept introduced over nearly four centuries ago.

At the turn of this century, Strogatz chronicled in his excellent book *SYNC* [17] the evolution of the synchronization concept and from which we freely draw with attribution. In the 1950s, the mathematician Norbert Wiener [18] identified the interaction of a spectrum of frequencies measured from the human brain using EEG time series as being the basis of human consciousness. However, although largely correct, his intuition did not anticipate the way in which the application of mathematics would be used. That distinction is credited to Art Winfree [19], a mathematical biologist who in the 1960s identified the fundamental nature of the nonlinear interactions of oscillators. Importantly, he showed that critical dynamics produced transitions from the disordered random behavior of microscopic degrees of freedom to highly ordered macroscopic degrees of freedom undergoing synchronous motion. He was thus able to identify dynamic self-organization as the mechanism underlying biomedical synchronization as in circadian rhythm, the entrainment of the pacemaker cells in the sinoatrial node of the beating heart, and elsewhere in the body’s physiological ONs. Strogatz credits Winfree with explaining that the resulting synchronization produced an alignment in time as distinct from the spatial alignment previously observed in physical phase transitions, e.g., the transition of a material from its fluid to its gas or solid phase.

A simplified oscillator model of self-organization in time was devised by Kuramoto [20] in the 1970s, which included the insights of both Wiener and Winfree, but with a symmetric interaction among the oscillator modes. The symmetry assumption enabled Kuramoto to obtain analytic solutions and thereby be the first to determine that a population of entities, from fireflies to brain cells, must have ‘sufficiently similar properties’ to synchronize their complex dynamics. While the individual oscillators in the Kuramoto model are regular, the emergence of global synchronization is independent of whether the individual oscillators are regular or stochastic.

The term ‘normal synchronization’ refers to the entrainment of the emergent dynamic global variables of two or more interacting networks. Consequently, this would include the critical dynamics of many-body phase transitions in the taxonomy of the expanding definition of synchronization. The individual dynamic networks in the process of chaos synchronization are chaotic and surprisingly do synchronize with other such networks while simultaneously maintaining the chaotic dynamics they had in isolation.

It was thought for a long time that chaos was incompatible with synchronization, that is until Pecora and Carroll [21] decided to apply chaos theory to encrypting messages in a chaotic signal for the purpose of communications. The ‘sufficiently similar properties’ are the fractal manifolds (attractors) of the sender and receiver. The chaotic fluctuations mask the message of the sender, which is retrievable using the deterministic dynamics of the second chaos generator identical to the first in the receiver. This strategy of driving a computer simulation of a receiver (a system with a strange attractor solution) with a chaotic signal transmitted from a duplicate of itself was indeed sufficient to coax the two into synchrony. Note that this kind of synchrony is quite different from the instantaneous time tracking of two deterministic trajectories, and both these forms of synchrony are separate and distinct from complexity synchronization. It also provides a rationale for complexity synchronization for two or more interacting ONs, that being the exchange of fractal information with the fractal dimension providing the coding.

Examples of this form of chaotic synchronization appear in chemical oscillations by means of a Belousov–Zhabotinsky reaction [22]; heat relaxation oscillator [23]; chaotic systems [24]; and control for chaotic systems [25]. The forerunner of these applications is discussed in the excellent text on the universality of synchronization in nonlinear science [26].

### 1.4. Rehabilitation and Complexity

We have defined complexity in terms of the fractal dimension of crucial event time series having a form of temporal complexity, and we have elsewhere proven that perfect synchronization results from interactions between the two complex networks, with the more complex network restoring the temporal complexity of the less complex network [27]. Quite generally, this restoration of the fractal dimension can be interpreted as a form of rehabilitation [4], an example of which is given by the therapeutic effect of arm-in-arm walking. Almurad et al. [28] demonstrated that if an aged patient walks in close harmony with a young companion, the ‘complexity matching effect’ results in the restoration of complexity in the gait of the elderly. Mahmoodi et al. [29] proved that scaling synchronization is a consequence of the fact that a crucial event time series has a μ index in the interval 2<μ<3 using the modified diffusion entropy analysis data processing technique (see Section 2 Methods). However, in the ‘complexity matching effect’, the level of complexity of the interacting ONs are often out of balance in a healthy individual, and the ON with the greater complexity drives the ON with the complexity deficit. The driver perturbs the index of the driven to a higher level, and when the complexity of the driven becomes equal to the driver, the maximal transfer of information occurs, and the two are in synchrony, with their fractal dimensions becoming stable and equal [7,28].

This notion of matching in the ‘complexity matching effect’ has developed into the idea of management resulting in the ‘Principle of Complexity Management’ [29] to include the influence of one ON on the other in the sense of an ensemble average for non-ergodic time series. On the other hand, complexity synchronization is realized through the scaling of single time series and occurs when the interaction between the two ONs is strong enough that transfers of information between the two change the driven ON’s statistics induced by the driver. But, surprisingly, the inverse power-law index of the driver is changed as well as that of the driven. Consequently, the scaling indices of the two ONs dance around a value which is between that of the driver and that of the driven [27], indicating that the two ONs have equal strength in transferring information among ONs in the more general case of healthy NoONs. This coordination of time series is observed experimentally among the triad of ONs [5], subsequently discussed (in Section 3 on Results).

## 2. Methods

Let us consider a useful way to characterize how the brain exchanges information with the two other major physiological ONs depicted in Figure 1. The three ONs whose time series are explicitly considered are the brain, heart and lungs ONs. Even a casual view of the typical ten-second time series shown along with a cartoon of the ON generating it reveals that they share no apparent features in common, much less the existence of any synchrony with one another. The brain’s EEG looks like a random signal; the heart’s ECG gives the impression of a two-state periodic oscillator, a normal sinus rhythm; and the normal breathing time-series pattern resembles the kind of nonlinear water wave observed heading in toward the shore and eventually crashing on the beach. Yet Mahmoodi et al. [5] showed that when these three time series are simultaneously measured, they in fact possess a remarkably new kind of synchrony in their normal healthy interactive state.

To demonstrate this new kind of synchronization, let us consider the ten-second time series for each of the three simultaneous measurements of the brain, heart and lungs depicted in Figure 1. The three time series depicted therein are denoted by $X_{j}\left(t\right)$ as well as the 63 other channels in the EEG measurement that are not shown in the figure. The subscript on the variable in this exemplar therefore denotes the output from channel *j* (=1, 2, 3) and the variable scales when the time *t* is multiplied by a constant λ resulting in $X_{j}\left(\lambda t\right)=\lambda^{\delta_{j}}X_{j}\left(t\right)$. Here, $\delta_{j}$ is the scaling index for channel *j*. Note that the scaling index is independent of time when the time series is a monofractal and is related to the fractal dimension of the neural network in the brain generating the time series in the vicinity of channel *j*. It is the scaling index $\delta_{j}$ of the empirical channel *j* crucial event time series, or equivalently the fractal dimension of the channel *j* time series $D_{j}=2-\delta_{j}$, that the modified diffusion entropy analysis technique enables us to find; see Table 1 for the many relations among the four parameters characterizing the time series.

The left panel of Figure 1 depicts the three time series to be processed using the modified diffusion entropy analysis technique. The results of this analysis of the time series on the left are depicted in the panel on the right, which anticipates the theoretical findings detailed in Section A.1. The empirical probability density function is obtained using the histograms from the diffusion argument and from which the Shannon/Wiener (SW) entropy for the three time series is constructed. Graphing the diffusion entropy versus the logarithm of the time yields a straight line with a positive slope. The results of the data processing indicate the existence of a deep structural relation common to these three very different looking time series.

The structural relation revealed by the modified diffusion entropy analysis data processing strongly suggests that the above homogeneous scaling be replaced with their average values, $\langle X_{j}\left(\lambda t\right)\rangle =\lambda^{\delta_{j}}\langle X_{j}\left(t\right)\rangle$, which means that the scaling is a property of the probability density function and not of the individual trajectories. In the next section, we examine the implications of the scaling probability density, one of which is that the empirical slopes in Figure 1 are, in fact, given by the three scaling indices $\delta_{j},$j=1,2,3. So, let us now examine the source of this remarkable result.

### Modified Diffusion Entropy Analysis (MDEA)

MDEA measures the complexity of the diffusion trajectory made by turning the empirical crucial event time series into a diffusion process. To avoid an unnecessary duplication of effort, we define the steps in the MDEA operating on a single heartbeat dataset but one significantly longer than the example just given. MDEA was applied to the post-processed continuous data from all 64 EEG channels, the electrocardiogram channel and the respiration channel of one participant in one session of neurofeedback training; for a more detailed description of the experimental protocol underlying the empirical dataset, see [5] as well as Section A.1.

For a stochastic process, the scaling equality is in terms of average values interpreted in terms of a scaling probability density function which can be written as [30]
(5)$$P\left(x,\tau \right)=\frac{1}{\tau^{\delta }}F\left(\frac{x}{\tau^{\delta }}\right),$$
where P(x,τ)dx is the probability that the random diffusion variate X(τ) is in the interval (x,x+dx) at time τ. In Section A.2, we show that the scaling probability density function is the general solution to a simple fractional kinetic equation [31,32], that is, a fractional equation in both time (having intrinsic memory) with an index α [4,30]
(6)$$D_{\tau }^{\alpha }\left[P\left(\tau \right)\right]=-\lambda^{\alpha }P\left(\tau \right),$$
and space (long-range inhomogeneity) with an index β. Equation (Equation 6) is a typical linear fractional rate equation with a solution given by the Mittag-Leffler function. The mathematical details from the fractional calculus are not of concern to us here; we note, however, that at early times the Mittag-Leffler function has the form of a stretched exponential, and asymptotically it becomes an inverse power law in time with an index α. We also note that the series expression for the Mittag-Leffler function has the analytic form of an exponential for α=1. Thus, the further α is from 1 (α<1), the slower the decay of the memory, which is to say the longer the intrinsic memory of the dynamic process reaches back in time.

The fractional equation which has the scaling probability density function as the renormalization group solution is a fractional kinetic equation, as we discuss in Section A.2 [30]. The scaling index δ is shown in Equation (A9) to be the ratio of the fractional derivative index at time α to the fractional derivative in space β, which is to say δ=α/β. The case (α,β) = (1,2) corresponds to a simple diffusion, with δ=1/2 having the fractal dimension D=1.5.

The scaling probability density function F(.) is unknown in general; however, for δ=0.5, it is Gaussian in the scaled variable $x/t^{0.5}$ and the process is diffusive. If the probability density function is Gaussian but δ≠0.5, the process is said to describe a form of anomalous diffusion called fractional Brownian motion by Mandelbrot and Van Ness [12], who first described it using fractional calculus. We note that fractional Brownian motion events are not of the renewal type because there is a long-term memory in the generating process, and therefore such a process cannot contain crucial events. Consequently, the more interesting case is when the unknown function is not Gaussian, but the statistics are of the renewal type and therefore cannot be fractional Brownian motion but can be either crucial events or non-crucial events, both of which can be of the renewal type, e.g., a Poisson process consists of non-crucial events that are of the renewal type.

You may have noticed that because crucial event time series are renewal they cannot have memory in the sense that fractional Brownian motion has memory, but these two kinds of memory can be distinguished by separately shuffling the two time series. A time series with a scaling index δ>0.5 and normal memory, such as fractional Brownian motion, when shuffled, will yield a scaling index of $\delta_{shuffled}=1/2$, whereas a time series consisting solely of crucial events with a scaling index δ>0.5 will, when shuffled, not change its scaling index $\delta_{shuffled}=\delta$. This counter-intuitive result was named ‘memory-beyond-memory’ by its discoverers Allegrini et al. [33] and is explained in varying levels of detail by West and Grigolini [4] and Bohara et al. [34], among others, but its existence has not been universally embraced by the theoretical community who study such things.

Thus, if the empirical probability density has the scaling form given by Equation (Equation 5), we note that a graph of the diffusion entropy ΔS(τ) versus the log of time (lnτ) makes it reasonable to interpret the slope of the empirical curve in Figure 1 to be given by the scaling index in Equation (A2) for the SW entropy. The three scaling indices indicate the values for the EEG, respiration and the ECG, for these simultaneously measured ten-second datasets make it also reasonable to interpret the brain to be the most complex, the heart the least complex and the lungs to have complexity between the other two members of the interacting triad. We emphasize that this ordering is not universal but does suggest that during this short time interval the part of the brain generating this signal was driving the other two ONs.

The results obtained for the three short datasets indicating that each of the time series has a constant fractal dimension entails that these time series are given by fractal scaling processes. If that were the end of the story it would still be a valuable result, but it would be a very restricted one because it would not allow for the ONs to adapt to new situations, which is an obvious capability of healthy ONs. The monofractal behavior of the three ONs observed for the ten-second time interval are due to intra-ON interactions that do not change their fractal behavior in this time interval because they have not been alerted to do so by any inter-ON information exchange and therefore they do not change their fractal dimensions. In the next section, we apply the modified diffusion entropy analysis to a significantly longer total time dataset for this triad of simultaneously measured time series and find they are substantially richer in information, and this longer total time reveals the ONs true nature, which is multifractal. In this way, each of the 66 ONs generates a separate multifractal but does so in a coordinated way under the influences of the other 65 ONs.

So, let us now examine how the monofractal behaviors of these 66 ONs are modified over longer times by the information exchange during their mutual interactions. In Section A.1, the modified diffusion entropy analysis data processing technique for multifractal crucial event time series is briefly reviewed using the electrocardiogram time series from the triad of measurement types as an exemplar. The results of this analysis applied to the 66 simultaneously measured time series are depicted in figure in Section 3.

## 3. Results

In this section, we present a new way to characterize how the brain exchanges information with two other major physiological ONs, those being the respiratory and cardiovascular ONs with the results of their interactions portrayed in Figure 2. This figure depicts a quasi-periodic time dependence of the scaling indices $\delta_{j},j=1,2,\ldots ,66$ for the processed datasets, as discussed in the Section 2 on Methods, from each of the 64 channels of a standard EEG, along with the those from the cardiovascular and respiratory ONs that were simultaneously measured. Note that the time scale is such that if we randomly select a point along the time axis and magnified the time series in the vicinity of that point, we would obtain something similar to the three constant fractal dimensions in Figure 1 but not necessarily as closely aligned as the results depicted therein. It is clear from the left panel of Figure 2 that the quasi-periodic behavior of the scaling indices from the EEG channels are in synchrony with those from the cardiovascular and respiratory ONs.

Figure 2 affords a clear answer to the following question: How does complexity synchronization occur in scaled metrics from empirical datasets of brain, cardiovascular and respiratory networks? There are 64 different EEG channels, corresponding to the green lines. For each of them, using stripes of a proper size was possible to find the scaling $\delta_{j},j=1,2,\ldots ,64$ in a sufficiently small bin of time Δt to define an ‘instantaneous’ value of $\delta_{j}\left(t\right),j=1,2,\ldots ,64$. This by itself is a significant benefit of using modified diffusion entropy analysis. The same method of analysis was applied to the respiration (red curve) and ECG (blue curve) time series. While the interaction between the brain and the physiology ONs of the body has a number of conjectured forms in the physiology literature, Figure 2 firmly establishes that the complexity of these different physiological processes as measured by their respective multifractal dimensions are synchronized.

The visual impression of the synchrony of the processed datasets in this last figure is borne by the cross-correlation coefficients of the three scaling index types recorded in the right-side panel being in the narrow interval [0.70, 0.73]. This synchrony of the multifractal behavior is a clear manifestation of the complexity synchronization phenomenon, which is not a strict deterministic phenomenon but is a statistical regularity.

Note that it is the scaling indices that are changing over time in Figure 2, thereby indicating the coordinated multifractal behavior of the time series from the brain, ECG and respiratory ONs. The neuroscientist Buzsaki [35] commented that transient coupling between various parts of the brain supports an information transfer to, and from, other ONs. We draw this to the readers’ attention because the scaling results shown in the figure support this conjecture. But a word of caution is appropriate here in that the synchrony observed in Figure 2 for the scaling indices does not necessarily have anything to do with the synchronous behavior observed from the central moment correlation properties of the time series observed in the insightful paper on the ‘complexity matching effect’ by Delgnieres et al. [36].

The quasi-periodic nature of the scaling parameter depicted in Figure 2 provides insight into the way the dynamic information from the brain, heart and lungs is exchanged during their mutual interactions. The gray curves in this panel depict the instantaneous scaling index over all 64 EEG channels, which is compared with the scaling index for the cardiovascular network (blue curve), the scaling index of the respiratory network (pink curve) and the ‘scaling index for the brain’ obtained by averaging over the 64 channels of the EEG (black curve). This figure indicates that all the ONs (or 66 network channels) have dramatic changes in complexity over time, being a direct consequence of their inter-ON and intra-ON interactions. This time dependence of the scaling indices means that the fractal dimensions become multifractal dimensions with quasi-periodic time dependencies. To properly interpret the behavior depicted in Figure 2 requires that we answer the following question: What is a scaling parameter and what does it entail regarding the underlying dynamic network? A question we partially answer in the Discussion Section.

## 4. Discussion

Healthy ONs being hosted by a healthy living network have equal strength in transferring information among ONs in a NoONs but give rise to time series whose fractal fluctuations contain control information that guides both the internal behavior and external information exchange of these complex ONs within the NoONs. In fact, the health of the human body is determined by the multifractal dimensions or alternatively by the scaling indices $\delta_{j}\left(t\right)$. The scaling index has the value 1 as the ideal condition for the health of the human body. It is a singular condition corresponding to the largest possible scaling and has been used in the analysis of heart rate variability (HRV) time series as a diagnostic indicator separating patients who have congestive heart failure from those that are healthy [34,37]. More generally, the brain may also explore the condition μ<2 and the condition μ>2, with a decrease in the scaling value.

It bears repeating that Figure 2 provides a clear description of how complexity synchronization occurs in the empirical datasets from the brain, cardiovascular and respiratory networks using modified diffusion entropy analysis. One of the significant benefits of using this data processing technique is that it reveals the crucial event character of these 66 channels or ONs. As mentioned, the matter as to how information is exchanged among the channels of the brain and the ONs of the body is presently a matter of scientific conjecture; however, Figure 2 proves that the complexities of these 66 different channels are synchronized. It is this synchrony of the multifractality of the interacting ONs that enables the efficient exchange of coded information among the ONs within the human body.

It is the fractal statistics of physiological fluctuations that determine the spatial properties of the tree-like branching structures of the human lung, arterial and venous systems and other ramified structures [38]. Statistical fractals also determine the waiting-time distribution of the time intervals successive beats of the human heart [4,33,39,40], in respiration [37], in dyadic conversation [41], in the human nervous system [42,43], in the dynamics of the brain [44,45], in the walking rehabilitation of the elderly [28,46], in motor control [36,47] and in interpersonal coordination [48,49], to name just a few. But it is worth quoting Buzsaki on the fractal nature of the brain [50]:

No matter what fraction of the brain web we are investigating, neuronal loops are the principle organization at nearly all levels. A physicist would call this multilevel, self-similar organization a fractal of loops.

The fractal paradigm is captured by the statistics of the scaling probability density function and is a consequence of the fact that the scaling probability density function is the solution to a fractional kinetic equation, as sketched out in Section A.2. The dominant characteristics of fractal statistics are spatial (*x*) inhomogeneity, temporal (*t*) intermittency and the phase-space trajectory (x;t) replacement of the dynamic variable X(t). In the phase space, the scaling of the dynamic variable is replaced by a scaling probability density function of the form given by Equation (Equation 5), which is true quite generally for ON statistical phenomena [51]. The first moment of X(t), using the scaling probability density function, recaptures the homogeneous scaling form of the dynamic variable, $\langle X(\lambda t)\rangle =\lambda^{\delta }\langle X(t)\rangle ,$ whose solution has the same power-law time dependence discussed in connection with the scaling relations recorded in Table 1. Such processes have monofractal statistical behavior.

So, what does it mean when we obtain a multifractal statistical process, which is to say a time-dependent scaling parameter $\delta \left(t\right)?$ The short answer is that the statistics are given by the scaling probability density function but with the constant scaling index replaced with the time-dependent scaling index. The longer answer is, well, longer, because it must provide an understanding for the time dependence.

Lloyd et al. [52] argue in their review that biological systems are homeodynamic (or homeorhetic) as a manifestation of an ON’s ability to self-organize at behavior bifurcation points where an ON loses stability and restabilizes in a new state. As a result of this self-organization, ONs display complex behaviors with a spectrum of emergent characteristics, including bistable switches, thresholds, mutual entrainment, as well as periodic behavior. These processes may proceed on different time scales, from very rapid processes at the molecular level to the enormously long time scales of evolutionary change; see, for example, Steven Gould’s long discussion on punctuated equilibrium theory in his 1400 page book *The Structure of Evolutionary Theory* [53]. It is apparently the dynamic self-organization under homeorhetic conditions that makes possible the organized complexity of life. Given the changeable behavior of the underlying complexity of NoONs, it is not surprising and is to be expected that the statistics are multifractal rather than monofractal in living networks.

The identification of fractal statistics was a major step away from the signal-plus-noise model that had dominated the engineer’s view of the disruptive role of fluctuations in complex phenomena. The scaling behavior of biomedical time series entails the fact that the fractal fluctuations are not normally disruptive but are rich in information that is exchanged in the interactions among ONs. The strength of the fractal paradigm lies in the fact that no single scale or frequency carries the signal, but rather pieces of the signal are encoded across a spectrum of scales. In this way, when noise does disrupt the signal, the repetitive nature of the fractal scaling ensures that, although the signal may be weakened, the information will not be totally lost. One way the resilience of a fractal structure to both internal and external normally disruptive fluctuations was understood involved using a fractal scaling model of the airways within mammalian lungs [54,55]. These mathematical results prompted the adoption of the interpretation that fractal structures are preadapted to such disruption, which meant that a fractal structure already possessed the scale being presented by the disputer, or a scale reasonably close to it. However, the modern term ‘resilience’ is more neutral with respect to a causal mechanism than is the more descriptive term ‘preadapted’.

However, even this generalization of the engineering paradigm was shown to be too restrictive to properly describe the richness in the statistics of physiologic time series. Most if not all physiologic time series are found to be characterized by time-dependent scaling parameters and therefore belong to the broader class of complex processes of multifractals. The time series from the heart, lungs and brain give some indication of the reasonableness of this interpretation. The scaling indices for the brain, heart and lungs have a range of variation (min:max) given by Δδ≈ 0.267 (brain); 0.122 (heart); and 0.121 (lungs). This increased flexibility of the range of variation in the brain’s multifractal scaling index may well be a reflection of the brain’s multi-task functionality, given the fact that the brain is itself a living NoON.

The health of the living NoONs that comprise the human body is determined by the scaling properties of the various ONs. It is the fractal scaling that determines how well the overall harmony is maintained, because the ideal health condition of the body is represented by δ=1. This singular condition corresponds to the largest value of the scaling index. The brain may also explore the conditions μ<2 and μ>2 with a decrease in the value of the scaling index δ. Consequently, it was recognized that disease and injury are described by the loss of variability (complexity) [54,56], and for that reason, the strategies used for combating disease/injury are being critically re-examined. For example, experiments show a preference in the response of physiologic ONs to 1/f-signals over that of white noise, indicating a sensitivity of these ONs to fractal scaling control [42,44,57]. In the more general rehabilitation context, the strategy determining how we develop life support equipment is another important example of the need for re-examination. The tradition in applying life support strategies is to supply blood at the average rate of the beating heart, to ventilate the lungs at their average rate and so on for the other ONs necessary for sustaining life [54,58].

It is clear from Figure 2 that the condition δ>0.5 is always true, namely, all the physiological activities of the body adopt a super-diffusion approach of different intensity. The parameter μ referring to the time distance between two consecutive crucial events remains smaller than μ=3, which is the border with the region μ>3 corresponding to the ordinary statistical behavior of thermodynamic equilibrium. The condition μ>3 entails Gaussian statistics for the time series, thereby losing the renewal statistics of the empirical time series. This loss of crucial event time-series status entails that the ON undergoing such a transition is either diseased or has been damaged by an external cause.

We must consider the way nature has resolved the difficult problems of providing robust methods for ONs to exchange information with one another, with information flowing from the more complex to the less complex network [7]. Then, that knowledge is applied to the least invasive kind of intervention necessary for recovery. The way ONs exchange information provides guidance on how medical devices ought to intervene to facilitate recovery from illness/injury through rehabilitation. The least invasive method of rehabilitation is one that uses an ON’s own strategies to establish the road from illness or injury back to health. The lungs respond best to the natural driving of fractal bio-ventilators and the heart to the fractal cardiopulmonary bypass bio-pumps, each driven by the appropriate spectrum of fractal bio-frequencies; see the experiments successfully carried out as well as the clinical successes of Mutch and colleagues [58,59,60].

## 5. Conclusions

We conclude that an ON’s emergent time series, whose fractal properties are determined by its scaling index, not by its detailed microscopic dynamics, determines the health of that ON and ultimately of the human body. This scaling codifies the success of that ON in carrying out its function. Moreover, the index quantifies the information shared with the other ONs within the NoONs. We draw this conclusion from what we have learned by processing the interacting time series from the triad of the brain, heart and lungs, whose overall health is determined by the information shared among the three over time [5]. Given this result, what can we further conclude about the universality of complexity synchronization?

A possible mechanism for the quasi-periodic complexity synchronization among the time series generated by the brain, heart and lungs was suggested by reading Buzsaki’s 2006 book. The thalamus, being a hub in cortical–thalamic network interactions, serves as an integration center through which ‘reciprocal causality’ exists among various brain regions and likely among the brain and various ONs [50]:

‘The thalamus is a large collection of relay nuclei, a kind of customs and border patrol agency. These nuclei are the only source of information for the neocortex about the body and the surrounding physical world … The principal mechanism of the cortical-thalamic-cortical flow of activity is self-sustained oscillations …’

The meaning derived from this quote is that the integration center is inherently oscillatory, so that the interaction of multiple networks is coordinated and oscillatory. The above insight was taken from Cycle 7 of Buzsaki’s *Rhythms of the Brain* in his discussion of resting and sleep states void of external sensory stimulation or motor activity, but we think it readily generalizes to waking and task states, although such states would be constrained or influenced by sensory input and motor output.

The statistical analysis of the 66 empirical ON time series, in fact, support the conclusion that ONs generate crucial event time series, a condition necessary for complexity synchronization. Moreover, we may also conclude that such crucial event time series have fractal statistics, which also facilitates complexity synchronization in multi-ON interactions. However, the fractal nature of these time series is not constant, which is to say they are not monofractal but change with the vagaries of the interactions of one ON with another, because other ONs are the environment in NoONs. These multifractal time series are produced by the internal dynamics of the individual ONs and can be described using Self-Organized Temporal Criticality models. The critical self-organization generated by an ON’s internal dynamics produces in time [27] what the Self-Organized Criticality model produces in space [61]. Consequently, physiologic phenomena are always multifractal and their spectral width is a measure of the state of health of that network [51,62,63,64] and consequently the overall health of the individual.

ONs use fractal statistics to preadapt their dynamics to potentially disruptive perturbations [54], whereas multifractality generalizes that adaptability to the breakup of classical trajectories into fractal trajectories with the onset of chaos [32]. This kind of adaptation enables going beyond what Taleb labeled ‘antifragile’ behavior [65]. The antifragility concept encompasses how things, in our case the ‘thing’ would be an ON time series, gain from disruption rather than being weakened by it. The increase in uncertainty that antifragility promotes in order for an ON to become stronger, i.e., increase its complexity, in the face of disruption and adversity, whether produced inside or outside the ON, is precisely what is measured by the width of the multifractal spectrum.

A remarkable aspect of multifractality is that it is not just a consequence of the critical dynamics of ONs, in which Self-Organized Temporal Criticality would be a reasonable driver for such behavior. The scaling behavior of these three physiologic time series is invisible to most data processing techniques and thereby so too are the crucial events. The hidden interdependence is above the level of time-series scaling generated by the interactions of the three ONs, those being the heart, lungs and brain. It is only after the modified diffusion entropy analysis processing of the time series that the complexity synchronization mechanism tying the three ON time series together is revealed.

Complexity synchronization is a newly identified evolutionary mechanism ‘Nature devised’ to enable NoONs to continue performing their global functioning behaviors by incorporating the complex dynamic feedback from the host NoON into the guest ON’s individual dynamics. The multifractal dimension indicates how information is encoded within ON time series and which guarantees a proper response regardless of the complexity of the healthy host state. The time-dependent fractal dimensional encoding insures efficient communication across multiple interacting ONs. The quasi-periodic oscillations are each statistically disrupted by distinct inverse power-law temporal frequency perturbations.

In summary, we conclude that a time series generated by the critical dynamics of a healthy ON is a homogeneous random fractal, with independent time intervals, and is consequently a crucial event time series. Furthermore, such a time series is modulated by a self-similar scaling index δ, giving rise to a fractal dimension $\mu_{D}=2-\delta$ that directly measures the complexity of the ON time series. The full complexity of an ON connected within a NoON is captured by the time dependence of the scaling index δ(t), resulting in a multifractal dimension of the interacting ON time series. We stress here that the quasi-periodicity of the three time series observed is necessary to carry out the distinct task in which the triad of ONs are engaged, and we expect even richer time dependencies to be revealed as the variety of tasks of differing levels of difficulty are carried out under controlled conditions or perhaps more coordination under conditions of complete rest or sleep.

We hypothesize that complexity synchronization is the mechanism necessary to coordinate the multiple time dependencies of the many interacting ONs composing NoONs, under changing conditions. Consequently, the degree of disruption of an ON’s time series produced by illness or injury may be quantified by the degree to which the complexity synchronization level among the ONs deviates from their value during their normal operation in a healthy NoON.

## Figures and Tables

**Figure 1 entropy-25-01393-f001:**
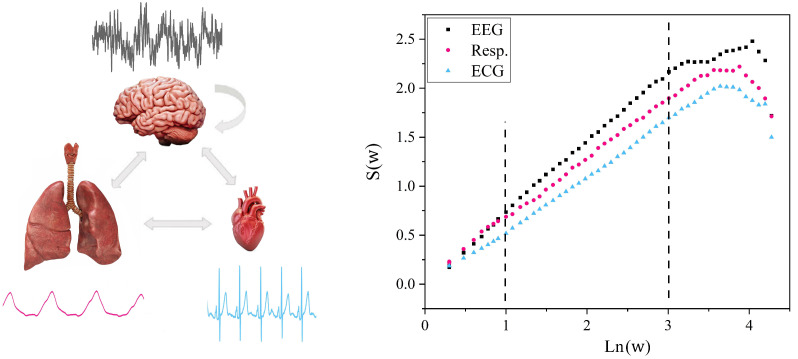
(**Left**) panel: The schematic depicts the three time series from the ONs of interest here, the brain, heart and lungs. Note that the three typical time series share no obvious common features. Also be aware that the information is exchanged simultaneously among all three as well as pairwise between the three ONs. Top is ten seconds of one channel of EEG time series; bottom left is ten seconds of respiration time series; bottom right is ten seconds of ECG time series; all three datasets are measured simultaneously. (**Right**) panel: The corresponding diffusion entropy analysis was used to process the diffusion random walks constructed from the three datasets depicted in the left panel (see Section A.1 for details or Mahmoodi et al. [5]). The entropy ΔS(w) is plotted versus the log of the time w as predicted in Section A.1 by Equation (A2) for a scaling probability density function, such that the three slopes between the dashed vertical lines yield the scaling indices for the corresponding time series. The slope is the measure of temporal complexity of the time series given by $\delta_{j}$, see Table 1. From Mahmoodi et al. [5] with permission.

**Figure 2 entropy-25-01393-f002:**
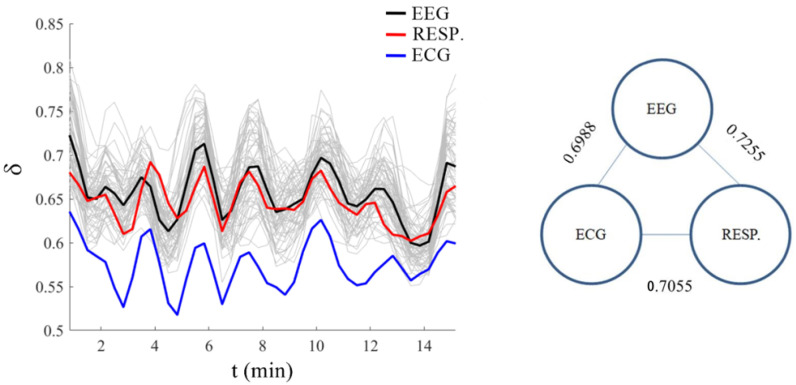
(**Left**) Panel: Light gray curves are the scaling indices $\delta_{j},j=1,\ldots ,64$ obtained by processing the 64 time series from the EEG channels and the black curve is the average over the 64 scaling indices at each point in time. The red and blue curves are the scaling indices obtained by processing the time series of the respiration and ECG channels, respectively. Modified diffusion entropy analysis processing was performed on each channel time series with stripe size of 0.01 for the ECG and respiratory data and 0.1 for the EEG data, using the jumping ahead rule, and on data windows of one-minute length of data (windows in increments of 20 s steps), respectively. The data were simultaneously collected while the participant was conducting the Go-NoGo shooting task (for details, see [5]). (**Right**) Panel: The corresponding pairwise cross-correlation coefficients (CCs) are calculated among the depicted EEGs channels, ECG and respiration scaling coefficients, with all calculated values of the three correlation coefficients falling within the interval 0.70 < CC < 0.73.

**Table 1 entropy-25-01393-t001:** This table makes easy reference to the scaling index δ from the above homogeneous scaling relation for the scaled variable X(t), relating it to the inverse power-law spectral density *S*${}_{p}$*(f)* index β through the waiting-time probability density function ψ*(t)* index μ in the two asymptotic regimes. The value μ=2 is the boundary between the underlying process having a finite (μ
>2) or an infinite (μ
<2) average waiting time and is also the point at which β
=1 where the process is that of true 1/f-noise. Consequently, β and μ are interchangeable measures of complexity. For an ergodic time series such as that determined by the waiting-time inverse power-law index, μ increases with decreasing scaling index δ and the complexity decreases. From [5] with permission.

	Scaled Functions	Parameter Relations	Parameter Range	
waiting-time PDF	$\psi \left(t\right)\propto t^{-\mu }$		1 ⩽ μ ⩽ 3	
power spectrum	$S\left(f\right)\propto f^{-\beta }$	μ=3−β		
scale variable	$X\left(t\right)\propto t^{\delta }$	μ=1+δ	1 ⩽ μ ⩽ 2	non-ergodic
		μ=1+1/δ	2 ⩽ μ ⩽ 3	ergodic
		δ=0.5	μ ≥ 3	

## Data Availability

All codes and the datasets are available at https://github.com/Korosh137/MDEA.git (accessed on 22 August 2023) or upon request to K.M.

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
