# Peer review of "Complexity Synchronization of Organ Networks"

_entropy, 2023, doi:10.3390/e25101393_

Round 1
Reviewer 1 Report
The manuscript titled 'Complexity Synchronization of Organ Networks' provides a historical background, definition, and review of a practical application of a concept the authors call 'complexity synchronization.' This concept is formalized into a set of analytical and theoretical tools which are explained and the use of which is demonstrated.
The concept of complexity synchronization is fascinating and compelling. The orientation to the concept provided by this manuscript will likely prove informative and impactful for readers. I will admit that found myself already wondering what relevance it has to my own world of scientific exploration and imagine that a high proportion of readers who invest in this manuscript will find the same. For these reasons I'm enthusiastic about this paper.
My enthusiasm, however, is dampened to the point of concern by two straightforward issues. The first, in short, is the issue of novelty. The second, in short, is the density of jargon.
Regarding the issue of novelty - While this manuscript in its present form has not been published, it is not clear what the present manuscript offers that hasn’t already been published. The extent of overlap with the Mahmoodi et al. Scientific Reports paper that is appropriately cited is very high. I can appreciate that the current manuscript may offer a bit more discussion of the background of complexity and synchronization as separate concepts, but the meat of both papers is largely indistinguishable to me. What is this manuscript distinct from the others?
Regarding the density of jargon – The current manuscript (and other manuscripts by the same authors) make abundant use of acronyms and extensive use of jargon. This is not against the rules, clearly. However, it created a major impediment to my understanding of the material. If the goal is to build greater awareness and adaptation of these ideas, the manuscript would be much improved to use the absolute fewest acronyms possible and to increase the sensitivity to what words are jargony.
Minor suggestions – the nature of the prose is inconsistent across the manuscript. In some places it is almost impenetrably technical whereas in other places it has a strong narrative flow whereas other places seem oddly casual. The second to last paragraph of the Discussion, (“Upon seeing this figure. . .”) is one such oddly casual one. It wasn’t clear what figure was being referenced and the comment seems apropo of nothing.
The English is generally high quality modulo my concerns above about excessive use of acronyms, jargon, and highly technical at times.
Author Response
I was pleased with the referee's thoughtful review of the manuscript and his alerting us to the two weaknesses in our presentation, those being novelty and density of jargon. It was the lead author's understanding that what had been presented in the two previous papers were project specific and therefore the medical significance of the results could have been easily overlooked. To remedy this short-coming we recast the paper in the form of a review. This change of orientation we believe enabled us to follow the referee's guidance and "use the absolute fewest acronyms possible and to increase the sensitivity to what words are jargony". Major changes are highlighted in yellow and are quite extensive. We deleted the paragraph of concern in the Discussion section. Not to justify, but to explain, the inconsistent nature of the prose was a consequence of having five strong personalities co-authoring the paper I was surprised you could not hear five distinct voices. We believe that particular problem has been solved by the rewriting being done by the lead author subject to approval of the other four.

Reviewer 2 Report
1. How does complexity synchronization (CS) occur in scaled metrics from empirical datasets of brain, cardiovascular, and respiratory networks?
2. Can the difference between the IPL index and the IPL index be used to quantify the relative complexity between interacting complex networks?
3. What is the importance of the scaling index in determining the health status of the organ network (ON) and indeed the human body?
4. Is there a clear synchronization or common feature observed between the brain, respiratory and cardiovascular systems and please elaborate?
5. How does the interpretation of the IPL index of the waiting time PDF differ for different anomalous diffusion scenarios, such as super-diffusion and sub-diffusion?
Some sentences need to be enhanced and pay attention to the grammar mistakes
Author Response
We thank this referee for her/his thoughtful questions. We have attached the immediate response by one author to each of his questions which is typical of the strong personalities involved. To answer the questions in the context of the paper we elected for one person to author the extensive revisions, with the final draft being approved by unanimous vote of the authors. The manuscript now reads more like a review, so that we requested change of category for the manuscript, and the answers to those appear in multiple places throughout the text. We hope the referee is satisfied with the response, which we answered by elaborating on the explanatory narrative. The major changes made are highlighted in yellow.

Round 2
Reviewer 2 Report
Most of the comments have been addressed. Pay attention to the specific contributions and their significance of the practical application.
Pay attention to some grammar mistakes.